# Combined Rotating Ultra-High-Resolution Spectral Domain OCT and Scheimpflug Imaging for In Vivo Corneal Optical Biopsy

**DOI:** 10.3390/diagnostics14131455

**Published:** 2024-07-08

**Authors:** Renato Ambrósio Jr., Louise Pellegrino G. Esporcatte, Karolyna Andrade de Carvalho, Marcella Q. Salomão, Amanda Luiza Pereira-Souza, Bernardo T. Lopes, Aydano P. Machado, Sebastian Marschall

**Affiliations:** 1Rio de Janeiro Corneal Tomography and Biomechanics Study Group, Rio de Janeiro 20520-050, Brazil; louiseesporcatte@gmail.com (L.P.G.E.); karol.acarvalho@hotmail.com (K.A.d.C.); marcella@barravisioncenter.com.br (M.Q.S.); blopesmed@gmail.com (B.T.L.); 2Instituto de Olhos Renato Ambrósio, Rio de Janeiro 20520-050, Brazil; amandalpsouza@gmail.com; 3Department of Ophthalmology, Federal University of São Paulo, São Paulo 04021-001, Brazil; 4Brazilian Artificial Intelligence Networking in Medicine—BrAIN, Rio de Janeiro 20520-050, Brazil; 5Brazilian Artificial Intelligence Networking in Medicine—BrAIN, Maceió 57072-900, Brazil; 6Department of Ophthalmology, Federal University the State of Rio de Janeiro (UNIRIO), Rio de Janeiro 20520-050, Brazil; 7Instituto Benjamin Constant, Rio de Janeiro 22290-255, Brazil; 8School of Engineering, University of Liverpool, Liverpool L69 3GH, UK; 9Computing Institute, Federal University of Alagoas, Maceió 57072-900, Brazil; 10Department of Research & Development, OCULUS, Optikgeräte GmbH, 35578 Wetzlar, Germany; s.marschall@oculus.de

**Keywords:** Pentacam^®^ Cornea OCT, spectral domain, cornea diseases, cornea ectasia

## Abstract

Purpose: This article introduces the Pentacam^®^ Cornea OCT (optical coherence tomography). This advanced corneal imaging system combines rotating ultra-high-resolution spectral domain OCT with sub- 2-micron axial resolution and Scheimpflug photography. The purpose of this study is to present the first experience with the instrument and its potential for corneal diagnostics, including optical biopsy. Methods: In this prospective study, the Pentacam^®^ Cornea OCT was used to image the corneas of seven patients. The novel wide-angle pericentric scan system enables optimal OCT imaging performance for the corneal layer structure over the entire width of the cornea, including the limbal regions. A detailed analysis of the resulting images assessed the synergism between the OCT and Scheimpflug photography. Results: The Pentacam^®^ Cornea OCT demonstrated significantly improved image resolution and ability to individualize corneal layers with high quality. There is a synergism between the OCT high-definition signal to individualize details on the cornea and Scheimpflug photography to detect and quantify corneal scattering. The noncontact exam was proven safe, user-friendly, and effective for enabling optical biopsy. Conclusions: Pentacam^®^ Cornea OCT is an advancement in corneal imaging technology. The ultra-high-resolution spectral domain OCT and Scheimpflug photography provide unprecedented detail and resolution, enabling optical biopsy and improving the understanding of corneal pathology. Further studies are necessary to compare and analyze the tomographic reconstructions of the cornea with the different wavelengths, which may provide helpful information for diagnosing and managing corneal diseases.

## 1. Introduction

Optical coherence tomography (OCT) has become crucial for diagnosing eye diseases and assisting surgical planning. It offers a fast, non-contact method to visualize the eye’s anterior and posterior segments [1,2]. Modern OCT systems provide detailed imaging of the human cornea through high axial and lateral resolution. Initially, OCT was primarily employed to assess the posterior segment of the eye. Over time, its utility for examining the cornea and anterior segment became recognized [3]. One of OCT’s primary uses in the anterior segment was to assess the flap created during laser vision correction (LVC), which remains widely performed today [4].

The development of refractive surgery has led to significant improvements in the imaging techniques used for the cornea and front part of the eye. If not detected, early stages of keratoconus (KC) and forme fruste of keratoconus (FFKC) are the main reasons for progressive iatrogenic ectasia following LASIK or other laser vision correction procedures [5,6]. Therefore, identifying this condition is a critical component of the preoperative evaluation for refractive surgery. A retrospective study has shown that nearly 15% of eyes that developed ectasia post-LVC had preoperative tomographic features resembling those without the complication [7]. Consequently, vigorous ongoing research aims to develop advanced tomographic technologies and software to detect early signs of ectatic diseases and assess the risk of ectasia progression [8]. Additionally, recognizing early forms of ectatic diseases has become increasingly important with the emergence of new treatments like corneal cross-linking and intracorneal ring segments, which are most effective when applied early [9,10,11].

Indeed, the introduction of Placido disk-based corneal topography, and its color-coded maps marked a significant technological breakthrough in the 1980s [12]. However, post-LASIK ectasia in patients with normal topographic patterns and no apparent risk factors highlighted the limitations of only assessing the cornea’s anterior surface [13]. This led to the adoption of three-dimensional (3D) corneal tomography, which provides a comprehensive view of the anterior and posterior corneal surfaces across the entire limbus and includes detailed pachymetry mapping [14]. Research has shown that this advanced technology is more effective at identifying the risk of ectasia in patients initially deemed suitable for laser vision correction than conventional topography. More recent developments in segmental tomography allow for the detailed examination of specific corneal structures, such as the epithelium, Bowman layer, and Descemet membrane [15,16]. Intriguingly, studies have observed that the corneal epithelium can remodel itself in reaction to irregularities in the underlying stroma, suggesting that a detailed analysis of the corneal epithelium could offer additional valuable insights for diagnosing KC [15,17,18,19].

The development of Scheimpflug imaging significantly advanced corneal tomography by enabling the measurement of corneal curvature and thickness from elevation B-scans. Numerous indices have been developed to facilitate early KC detection using this technology [20]. Nevertheless, there was a recognized clinical demand for a system capable of evaluating all these indices simultaneously. Artificial intelligence (AI) has addressed this need by providing sophisticated algorithms integrating various tomographic indexes, such as curvature, thickness, and aberrations [20]. Additionally, the analysis of epithelial thickness distribution has proven effective in identifying KC and monitoring corneal changes following LVC [17]. In cases of KC, the epithelium often conceals the true steepness of the underlying corneal stroma [21]. OCT has met the need for a comprehensive tomography tool capable of imaging the inner stromal layers with high axial resolution [21].

### 1.1. Scheimpflug Imaging

The Scheimpflug principle, a fundamental concept in photography, was initially outlined by Jules Carpentier in 1901 [22] and later acknowledged by Theodor Scheimpflug in his 1904 patent [23,24]. This principle involves arranging three theoretical planes—the film, the lens, and the plane of sharp focus—in a non-parallel configuration. Unlike conventional cameras, where these planes are parallel, in Scheimpflug’s technique, the lens is tilted so that the lens plane intersects both the film plane and the plane of focus, creating what is known as the Scheimpflug intersection. This alignment expands the focus depth and enhances objects’ sharpness at varying distances, with minimal distortion in the resulting image [24].

Rotating Scheimpflug was introduced in the Pentacam (OCULUS Optikgeräte GmbH, Wetzlar, Germany). It has a single rotating camera alongside a static one for frontal eye alignment and high-quality detail. The Scheimpflug camera captures high-resolution images of the anterior and posterior corneal surfaces, anterior iris, and lens.

### 1.2. Optical Coherence Tomography

OCT operates on the principle of low-coherence interferometry, which involves analyzing the time delay of infrared light reflected from structures within the anterior segment [1]. In practical terms, two OCT technologies, commonly Time-domain OCT (TD-OCT) and Fourier-domain OCT, are used for anterior segment imaging. Fourier-Domain OCT is divided into two main subtypes: Spectral-Domain OCT (SD-OCT) and Swept-Source OCT.

TD-OCT operates by employing a low-coherence light source that emits light, splitting it into a reference beam and a sample beam through a beam splitter. Scanning the reference beam path length with a moving mirror measures the amount of light reflected at different tissue depths along the sample beam. This depth scan is called an A-scan. By moving the sample beam across the object (e.g. a patient’s eye) and acquiring A-scans at various lateral positions, a 2D image is constructed—which is called a B-scan. A number of B-scans can be assembled into a 3D data volume.

In the initial stages, anterior segment OCT (AS-OCT) devices like the Zeiss Visante (Carl Zeiss Meditec, Dublin, CA, USA) and Heidelberg SL-OCT (Heidelberg Engineering, Heidelberg, Germany) utilized TD-OCT technology alongside 1310 nm superluminescent diodes. This setup enabled penetration depths of approximately 7 mm. However, these early devices suffered from slow operation and prolonged image acquisition times, managing only 2000 and 200 scans per second [25].

Fourier-domain OCT follows a similar setup as TD-OCT but with a critical difference: The reference mirror remains stationary, and interference between the reference and sample arms is detected as a spectrum [25,26]. This enables drastically higher A-scan rates and, thus, faster image acquisition. SD-OCT systems employ broadband light sources for illumination (typically superluminescent diodes) and detect each wavelength separately with a spectrometer. In Swept-Source OCT, a rapidly tunable laser illuminates the target consecutively with a broad spectrum of wavelengths. Both variants have unique strengths; hence, the choice of technology depends strongly on the application. Generally, Swept-Source OCT enables ultra-fast image acquisition, a very long scanning depth range, or deep penetration into scattering tissue [27,28,29]. In contrast, Spectral-Domain OCT excels when ultra-high axial resolution is required [30,31].

More recently, Fourier-domain OCT devices utilizing wavelengths between 820 and 880 nm have been modified for anterior segment imaging. Examples include the Zeiss Cirrus (Carl Zeiss Meditec, Dublin, CA, USA), Heidelberg Spectralis (Heidelberg Engineering, Heidelberg, Germany), Optovue iVue and RTVue (Optovue, Fremont, CA, USA), and Nidek RS 3000 (Nidek, Fremont, CA, USA). These devices typically offer 2–3 mm scan depths and speeds ranging from 26.000 to 110.000 scans per second. Widely used in modern ophthalmology practices, they excel at high-resolution imaging of specific sections of the anterior chamber angle. However, these devices require specialized lens adapters to capture images across the entire anterior chamber in a single scan, as they are primarily designed for posterior segment imaging [25]. In contrast, the MS-39 (CSO, Florence, Italy) is a dedicated anterior-segment instrument combining SD-OCT with Placido disc topography. With a scan depth of 8 mm, it can capture the anterior chamber in a single B-scan with an axial resolution of 3.6 μm.

There are several Swept-Source OCT systems available for anterior segment imaging. The Casia SS-1000 (Tomey Corporation, Nagoya, Japan) utilizes a 1310 nm light source. It offers comprehensive imaging coverage of 16 × 16 mm and can scan up to 6 mm in depth at 30.000 scans per second. The axial resolution is 10 μm, with a transverse resolution of 30 μm in tissue. Another specialized SS-OCT instrument for the anterior segment is the Anterion (Heidelberg Engineering, Heidelberg, Germany). It operates at wavelengths around 1300 nm and acquires 50.000 A-scans per second. With a scan depth of 14 mm, it can capture images from the apex of the cornea to the back of the crystalline lens with an axial resolution below 10 μm. The Triton (Topcon Medical Systems, Oakland, NJ, USA) was initially designed for posterior segment imaging but included an anterior segment adaptor. It employs a 1050 nm light source with a scan rate of 100 kHz and a scanning depth of 3 mm [25].

### 1.3. The Pentacam^®^ Cornea OCT

The current study utilizes a novel ophthalmic diagnostic instrument from the Pentacam^®^ family (OCULUS Optikgeräte GmbH, Wetzlar, Germany). The Pentacam^®^ Cornea OCT combines the proven Scheimpflug tomography system with an ultra-high-resolution spectral domain OCT system (Figure 1). The SD-OCT engine has a capture speed of 50.000 A-scans per second, operates in the 800 nm wavelength band, and features an axial resolution below 1.9 µm in corneal tissue.

The novel wide-angle pericentric scan optics perform a one-dimensional lateral scan of the OCT probe beam, aligning it near-perpendicular to the cornea surface over the entire width of the cornea and both limbal regions. This addresses two fundamental challenges with ultra-high-resolution OCT imaging of the cornea. Firstly, when designing an SD-OCT or Swept-Source OCT system, one must always trade off the axial resolution (i.e., depth resolution along the probing beam) versus the maximum scanning depth (Figure 2A). Therefore, state-of-the-art SD-OCT systems can offer ultra-high axial resolution but only with a limited scan depth. The SD-OCT scan depth of the Pentacam^®^ Cornea OCT is approximately 3.5 mm. This would be insufficient for imaging the entire cornea with a traditional telecentric scanning system, which keeps the probing beam parallel to the eye’s visual axis. The second disadvantage of telecentric systems is the increasing angle of incidence between the beam and the corneal surface towards the cornea’s periphery. While OCT allows for an axial resolution (along the beam axis) down to a few micrometers, the lateral resolution (perpendicular to the beam axis) is typically not finer than 20 to 30 μm if a large lateral field of view is required. Hence, fine details of corneal layers can only be resolved clearly at small angles of incidence.

The pericentric scanning optics of the Pentacam^®^ Cornea OCT (Figure 2B) keep the probing beam near-perpendicular to the corneal surface from limbus to limbus, enabling optimal use of the SD-OCT’s ultra-high axial resolution for visualizing the corneal layers. The resulting measurement window curves around the cornea with a lateral extent of 17 mm.

The Pentacam^®^ Cornea OCT synchronously acquires 25 OCT B-Scans and 25 Scheimpflug images in the same image cross-sectional planes. This measurement takes 1.1 seconds. In the 3D scan mode, the probe head rotates around the eye’s axis during the measurement, thus providing 25 OCT and Scheimpflug images at distinct angles for creating a 3D cornea model and en-face maps of important corneal parameters. The ultra-high axial OCT resolution enables measurement of the corneal epithelial thickness with micrometer-scale accuracy and provides thickness maps with up to 10 mm coverage in diameter.

The combination of Scheimpflug technology and SD-OCT offers unique synergistic effects. Scheimpflug technology using blue light is particularly sensitive to all eye structures responsible for scattering light due to the strong wavelength dependence of light scattering in corneal tissue [32,33]. Consequently, opacities in the crystalline lens or the cornea can be detected early through Scheimpflug imaging. On the other hand, the near-infrared light utilized in OCT technology passes through these structures mostly undisturbed, often making these light-scattering structures not readily discernible at first glance. However, OCT offers superior spatial resolution for visualizing fine details of light-scattering structures within the cornea.

Another approach of combining OCT and Scheimpflug corneal imaging in one instrument was demonstrated previously, where the near-infrared OCT probing beam also served as the illumination source for two Scheimpflug cameras located on either side of the OCT scanning plane [34,35]. In this method, both systems inherently acquire images from the same imaging plane. For a 3-dimensional measurement, the OCT scanning plane shifts relative to the cameras, which simultaneously adjust their focal planes using tunable objective lenses.

In contrast, the blue LEDs (475 nm UV-free) Scheimpflug imaging geometry remains fixed in all instruments of the Pentacam^®^ family, which perform the 3D scan by rotating the instrument head. Therefore, the imaging is performed with a fixed-focus camera objective. In the Pentacam^®^ Cornea OCT, the Scheimpflug camera employs its dedicated slit lamp projector, which ensures highly even illumination of the imaging plane. At the same time, using blue light maximizes the sensitivity for detecting light-scattering structures. The slit lamp projector and the OCT scanning system are precisely aligned to capture images at the exact location. This combination of technologies now offers the opportunity to detect scattering in the cornea at first glance and to analyze the affected regions in detail through OCT.

## 2. Materials and Methods

This retrospective, analytical, and non-interventionist study comprised seven patients with corneal changes. All participants underwent a comprehensive ophthalmic evaluation, including uncorrected distance visual acuity (UCVA), best-corrected distance visual acuity (CDVA), spherical and cylinder refraction, and Pentacam^®^ Cornea OCT.

The Ethics Committee approved the proposed investigation at the Federal University of São Paulo/UNIFESP/SP 2020 (# 4.050.934) and the Tenets of the Helsinki.

## 3. Results

We describe seven cases of patients with corneal abnormalities, including Cogan dystrophy, post-COVID-19 corneal fibrosis, Fuchs’ endothelial dystrophy associated with KC, superficialization of an intracorneal ring segment (ICRS), lattice corneal dystrophy, pressure-induced stromal keratitis (PISK), and Salzmann’s nodular degeneration.

A 65-year-old female patient was diagnosed with Cogan dystrophy with uncorrected visual acuity (UCVA) of 20/30 in the right eye (OD) and 20/150 in the left eye (OS) and with distance-corrected visual acuity (DCVA) of 20/25 in the OD and 20/40 in the OS. Figure 3 shows the OS biomicroscopy with classical epithelium changes, and Figure 4 shows Pentacam OCT images (Scheimpflug, the actual shape of the cornea, and the cornea zoom, which corresponds to axial compaction) with abnormal epithelium layers.

A 58-year-old male patient was referred for evaluation of corneal fibrosis in the OS (Figure 5) that began two years ago after a COVID-19 infection. In 2012, he was submitted for a photorefractive keratectomy (PRK) for moderate myopia in both eyes. His UCVA was 20/60 in the OD and 20/50 in the OS, and the DCVA was 20/20 (−1.00/−2.00 × 75) in the OD and 20/40 (−0.50/−2.00 × 65) in the OS. Figure 5 demonstrates the depth of fibrosis in the Pentacam OCT of OS. A phototherapeutic keratectomy (PTK) was indicated in the OS, and his DCVA improved to 20/20 with −0.75/−0.75 × 128.

Figure 6 shows the OS from a 67-year-old male patient diagnosed with advanced Fuchs dystrophy and KC in OS. He was already submitted to a cornea transplantation and phacoemulsification in the OD. His UCVA was 20/70 in both eyes, and his DCVA was 20/30 (−2.25/−2.00 × 15) and 20/50 (−1.00/−1.25 × 75) in the OS. The Figure 7 shows his OD Slit-lamp biomicroscopy. His Pentacam OCT is demonstrated into Figure 8 (OD) and Figure 9 (OS).

Another interesting case concerns a 35-year-old female patient who was referred for evaluation of a possible ICRS extrusion. Her UCVA was 20/200 in OD and 20/250 in OS, and the DCVA was 20/60 in both eyes. In the Scheimpflug image, the ICRS appears very superficial; however, on the Pentacam OCT, we can see that there is still an extensive layer of tissue over the ring (Figure 10).

The following patient is a 40-year-old male with lattice corneal dystrophy, a genetic eye disorder characterized by the accumulation of amyloid deposits in the cornea, leading to a gradual decline in vision. His UVCA was 20/50 in OD and 20/25 in OS; his DVCA was 20/25 (+1.25/−3.25 × 145) in OD and 20/20 (+0.25/−2.50 × 50). Figure 11 shows the biomicroscopy and the Pentacam OCT changes of this case.

Pressure-induced stromal keratitis (PISK) typically develops weeks to months after LASIK. This complication is associated with elevated intraocular pressures (PIO) and tends to worsen with steroid treatment. It occurs in the interface layer between the corneal flap and the stromal bed, leading to haze and potentially affecting vision if not managed appropriately. Figure 12A and Figure 13A show the Pentacam OCT of a 42-year-old man complaining about poor vision two months after bilateral LASIK of OD and OS, respectively. His UCVA was 20/200 in OD and 20/250 in OS, with a PIO of 44 mmHg in OD and 45 mmHg in OS. After two hours of administering carbonic anhydrase inhibitor orally, the PIO was 14 mmHg in both eyes, and we can observe the corneal changes, which deflate and uncurve in the Pentacam OCT (Figure 12B and Figure 13B). After three days, the PIO was 12 mmHg in both eyes, and there were no pathological corneal changes in Pentacam OCT (Figure 12C and Figure 13C) and the Scheimpflug image (Figure 14).

The following case concerns a 65-year-old male patient referred for a corneal pannus evaluation. He had a pterygium excision surgery 45 years ago and is now complaining about visual loss. His UCVA was 20/150 in both eyes, and his DCVA was 20/50 (+2.00/−0.50 × 125) in OD and 20/70 (+1.00) in OS. In the OS biomicroscopy, we can observe a Salzmann nodular degeneration better demonstrated in the Pentacam OCT. He was submitted to manual removal with PTK smoothing and mitomycin use, and one day after the procedure, his UCVA was improved to 20/80; the changes are shown in biomicroscopy (Figure 15A,B) and Pentacam OCT in Figure 16.

## 4. Discussion

We aim to explore and assess the synergistic effects in imaging with Scheimpflug and SD-OCT. Future work should explore tomographic measurements, focusing on the 3D reconstruction of elevation, curvature, and thickness data across the entire cornea and its layers, including layer analysis mapping the epithelium, Bowman’s layer, and the Descemet–endothelial complex.

Xialorain Li and col. developed a novel combined OCT and Scheimpflug imaging with simultaneous acquisition using the same near-infrared illuminating light [34]. However, it differs from the Pentacam Cornea OCT. Their OCT images provided high axial resolution (2.7 μm in the cornea). In contrast, the Scheimpflug image gave a significant depth of 12.5 mm and a large field of view (16.8 mm × 12.5 mm) image overview of the anterior chamber. Another difference was that they only imaged porcine and bovine eyes ex vivo, and the Pentacam Cornea OCT is clinically available.

Multimodal imaging is crucial in the comprehensive assessment of ectatic corneal diseases (ECDs), encompassing diagnosis, classification, staging, prognosis, treatment planning, and monitoring. A key goal of pre-refractive surgery evaluation is to assess ectasia risk, which involves identifying not only those with mild ectasia but also understanding the cornea’s inherent susceptibility to biomechanical failure to prevent keratectasia [10]. Accurately quantifying ectasia risk requires a thorough understanding of the corneal structure and how it interacts with surgical interventions. Significant advancements in corneal imaging have been made over the past 30 years, and this progress is expected to continue.

Several studies confirmed the hypothesis that artificial intelligence (AI) can be enhanced to improve ectasia detection [20,36]. Such advancements could increase the reliability of clinical decisions concerning screening individuals at risk for ectasia following LVC. AI is quickly becoming more relevant across various subfields of ophthalmology.

## 5. Conclusions

The Pentacam^®^ Corneal OCT represents a significant advancement in corneal imaging technology. Combining ultra-high-definition spectral domain OCT with Scheimpflug photography delivers exceptional detail and resolution, enabling optical biopsy and enhancing the understanding of corneal pathologies. Further research is needed to compare and analyze tomographic reconstructions of the cornea using different wavelengths, which could yield valuable insights for diagnosing and managing corneal diseases.

Future advances in AI should involve collecting comprehensive data from large populations and incorporating features from multimodal imaging technologies, such as corneal tomography, biomechanical properties, detailed analyses of the epithelium, Bowman’s layer tomography, axial length, ocular wavefront, as well as emerging fields like molecular biology and genetics.

## Figures and Tables

**Figure 1 diagnostics-14-01455-f001:**
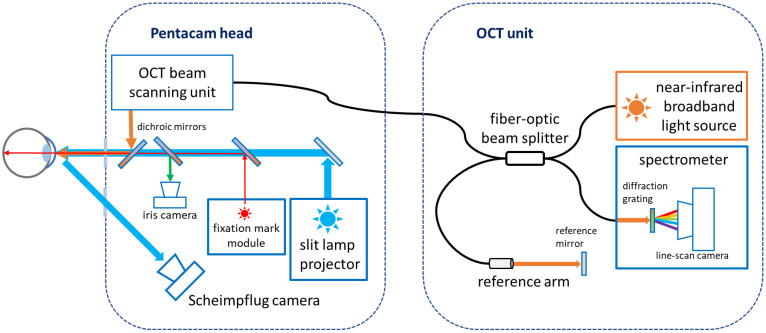
Optical system layout of the Pentacam^®^ Cornea OCT. Near-infrared light from the OCT unit is transmitted to the Pentacam head through a fiber-optic cable. After passing the beam scanning unit, the OCT probing beam is combined with the blue slit lamp illumination for Scheimpflug imaging. Both systems illuminate the same cross-sectional plane in the cornea.

**Figure 2 diagnostics-14-01455-f002:**
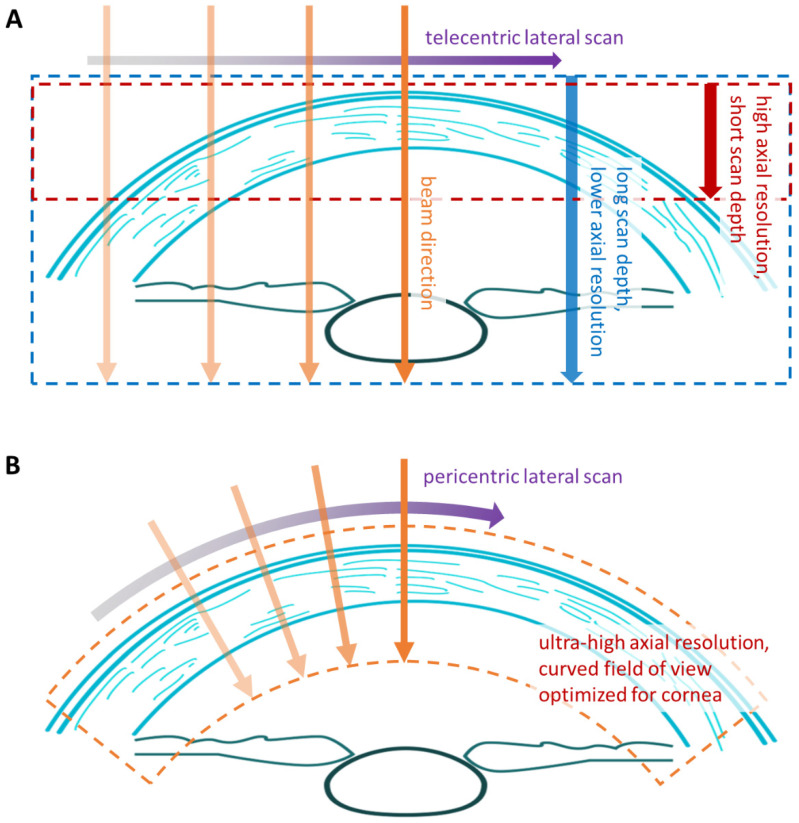
(**A**) SD-OCT or SS-OCT systems with telecentric beam scanning optics can either have an extended scan depth for imaging the entire cornea at moderate axial resolution or they can have high axial resolution with a limited scan depth. (**B**) The wide-angle pericentric scanning system of the Pentacam^®^ Cornea OCT keeps the probing beam near-perpendicular to the corneal surface. It creates a measurement window adapted to the corneal curvature. This way, the SD-OCT engine’s ultra-high axial resolution is optimally exploited to resolve the details of the corneal layer structure.

**Figure 3 diagnostics-14-01455-f003:**
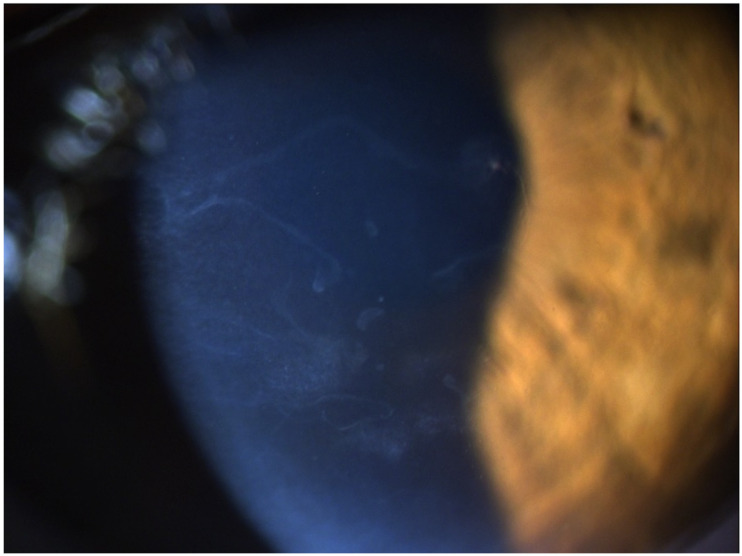
Slit-lamp biomicroscopy of classical epithelium changes in the Cogan dystrophy.

**Figure 4 diagnostics-14-01455-f004:**
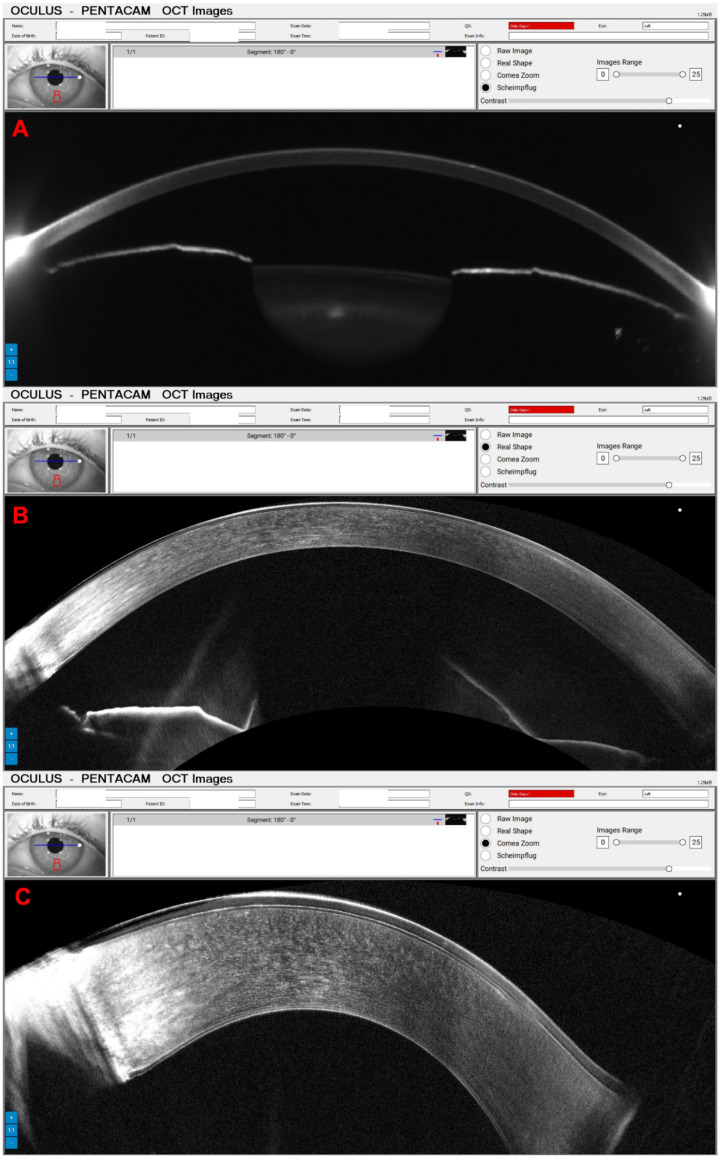
Pentacam OCT showing Scheimpflug image (**A**), the real shape of the cornea (**B**), and the cornea zoom (**C**) demonstrating abnormal epithelium in Cogan dystrophy (same patient of Figure 3). In the Cornea Zoom view, the cornea’s thickness is stretched to visualize all fine anatomical details across the entire width in one image.

**Figure 5 diagnostics-14-01455-f005:**
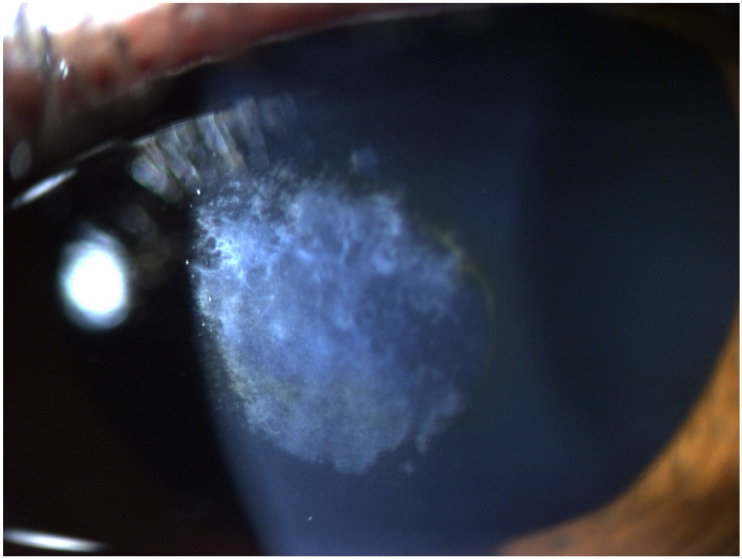
Slit-lamp biomicroscopy shows corneal fibrosis in the OS in a patient submitted to PRK in 2012 with a recent COVID-19 infection.

**Figure 6 diagnostics-14-01455-f006:**
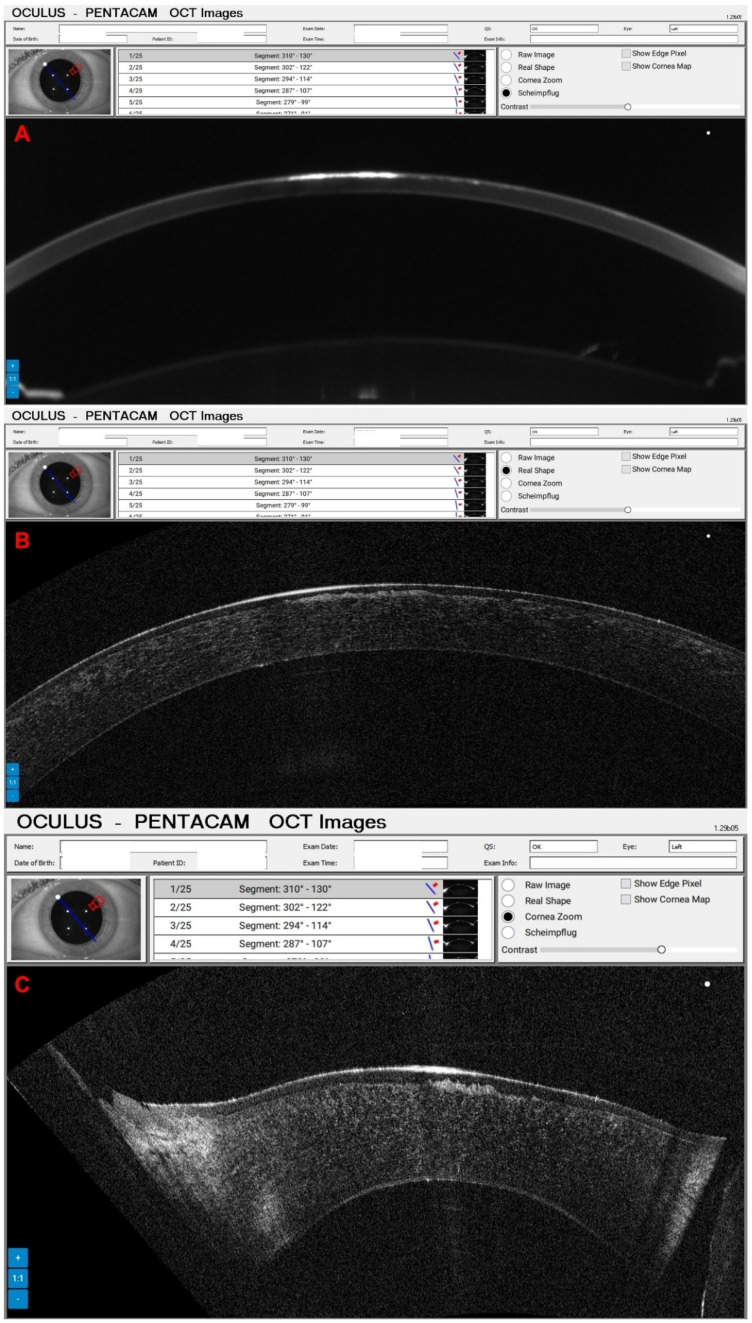
Pentacam OCT showing Scheimpflug image (**A**), the actual shape of the cornea (**B**), and the cornea zoom (**C**) demonstrating the depth of fibrosis in the same patient as in Figure 5.

**Figure 7 diagnostics-14-01455-f007:**
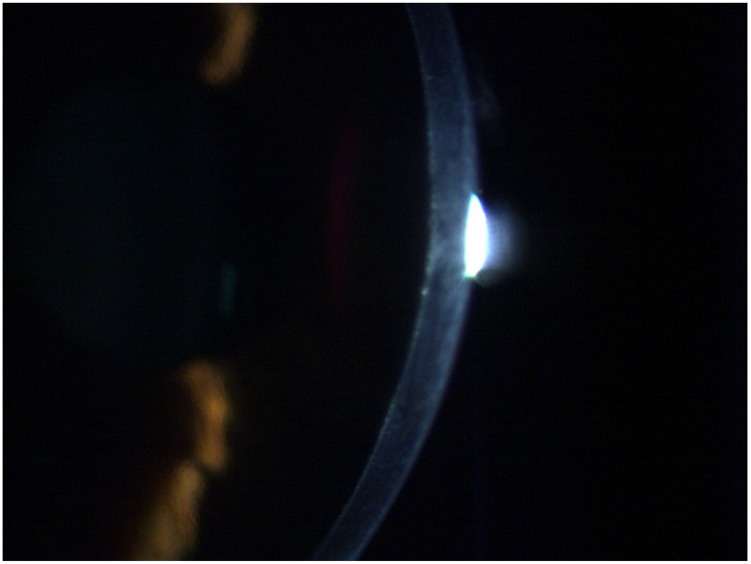
Slit-lamp biomicroscopy advanced of a 67-year-old patient with Fuchs dystrophy and KC in OS.

**Figure 8 diagnostics-14-01455-f008:**
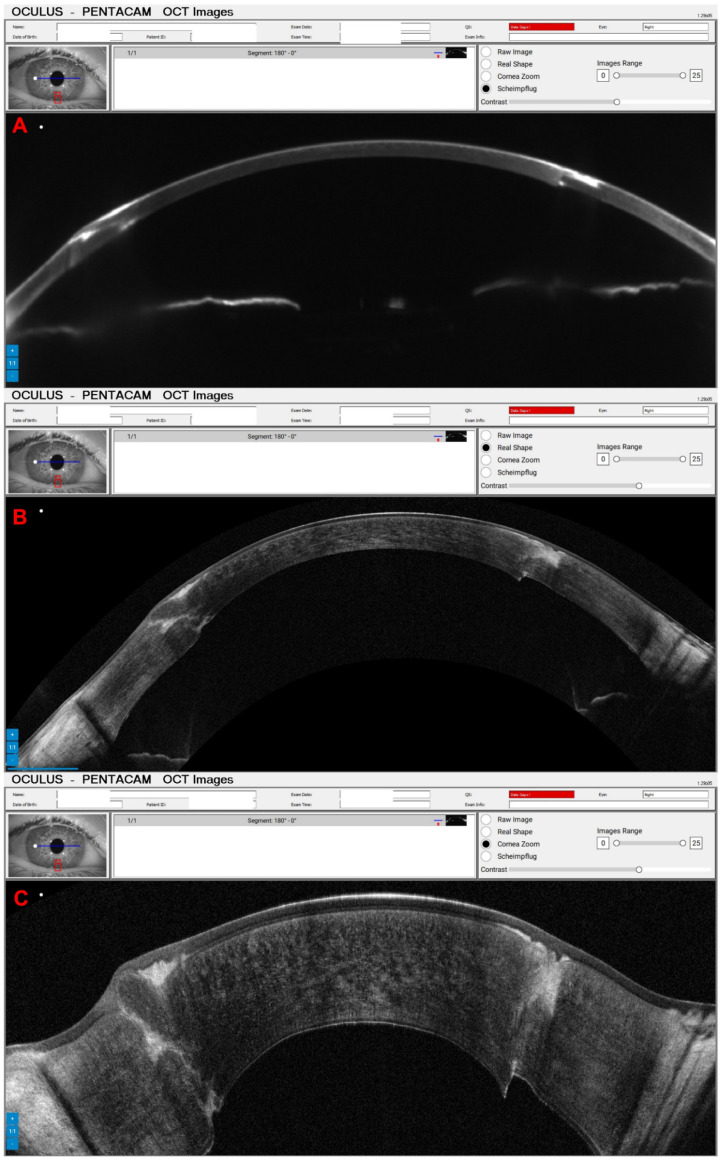
Right-eye Pentacam OCT of the same patient as Figure 7 shows the corneal transplant in Scheimpflug image (**A**), the real shape of the cornea (**B**), and the cornea zoom (**C**).

**Figure 9 diagnostics-14-01455-f009:**
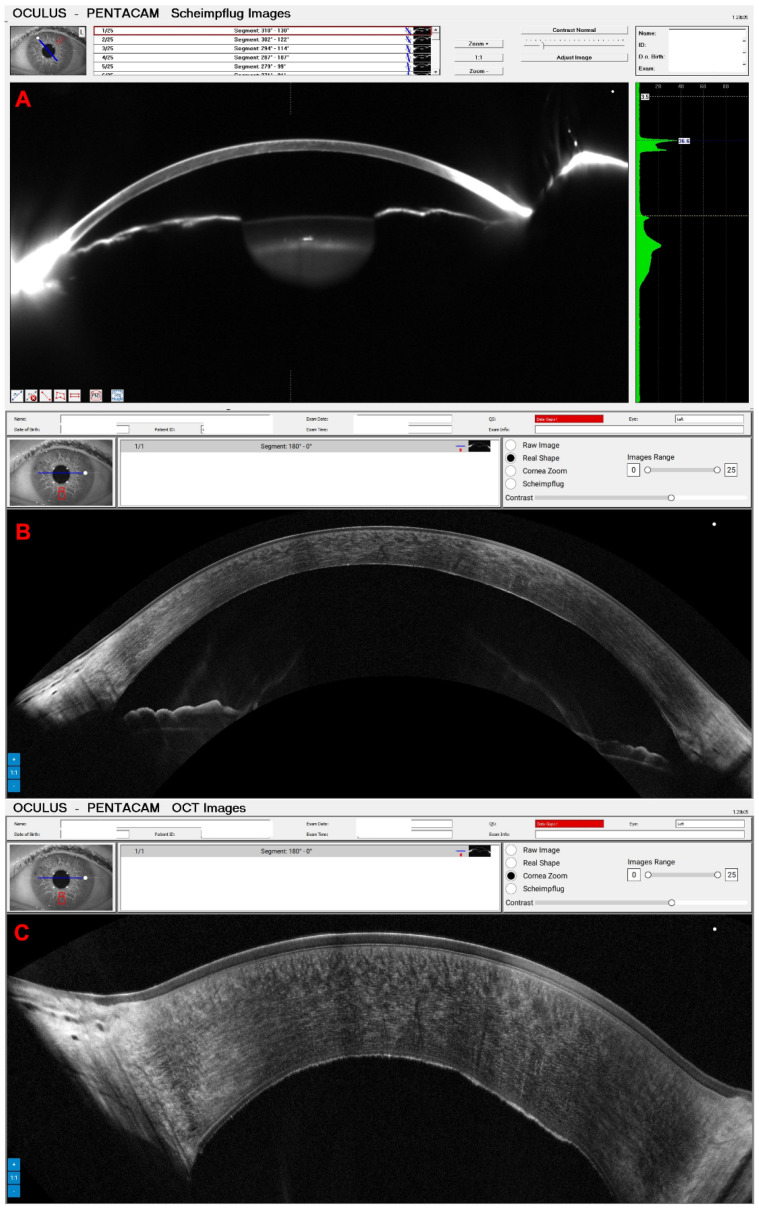
Left-eye Pentacam OCT of the same patient as Figure 7 and Figure 8 shows endothelium changes in Scheimpflug image (**A**), the real shape of the cornea (**B**), and the cornea zoom (**C**).

**Figure 10 diagnostics-14-01455-f010:**
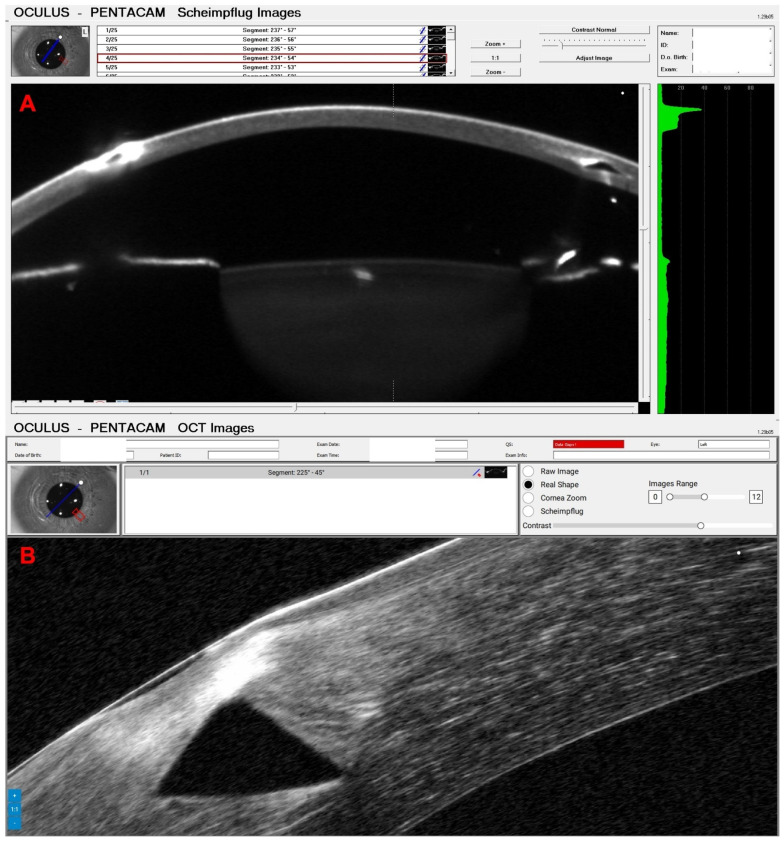
Pentacam OCT shows that the intracorneal ring (ICRS) is very superficial in the Scheimpflug image (**A**); however, in the OCT real-shape image (**B**), we can see that there is still a more significant layer of tissue over the ring.

**Figure 11 diagnostics-14-01455-f011:**
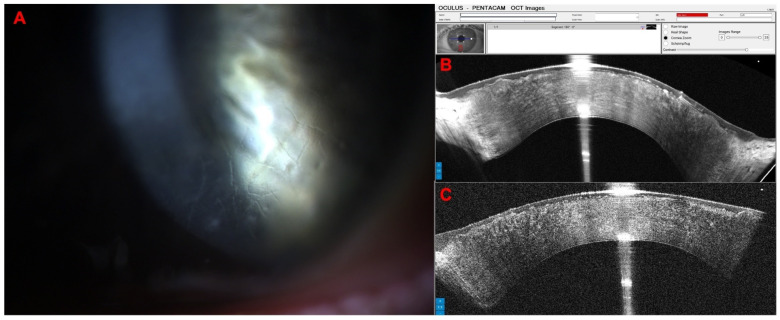
Slit-lamp biomicroscopy shows corneal lattice dystrophy (**A**), the Pentacam OCT cornea zoom (**B**), and real shape (**C**) with abnormal corneal stroma.

**Figure 12 diagnostics-14-01455-f012:**
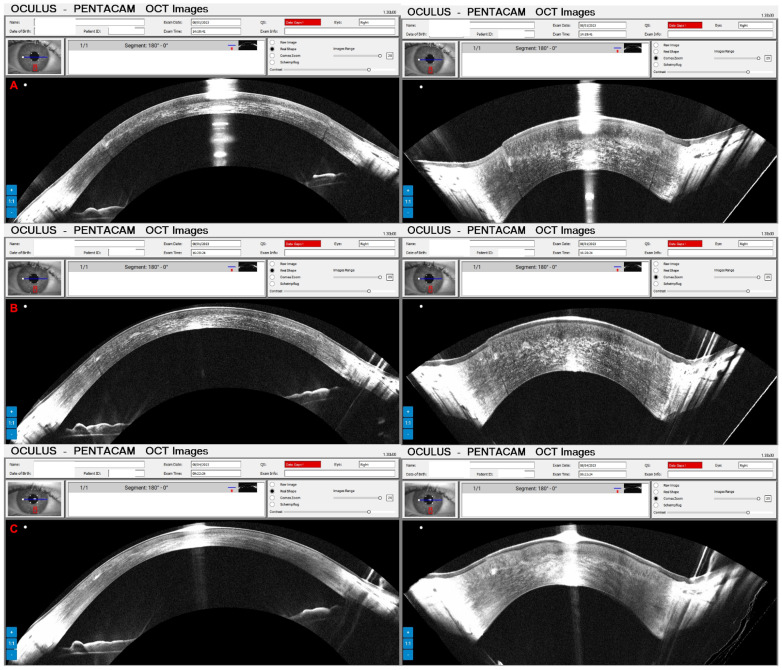
The OD Pentacam OCT of a patient with pressure-induced stromal keratitis (PISK) who underwent bilateral LASIK two months ago. In (**A**), at the first exam, the PIO of 44 and 45 mmHg in OD and OS showed a curved cornea with edema. In (**B**), after two hours of administering carbonic anhydrase inhibitor orally, we can observe the corneal changes, which deflate and uncurve; in (**C**), three days later, with PIO of 12 mmHg in both eyes and no pathological corneal changes.

**Figure 13 diagnostics-14-01455-f013:**
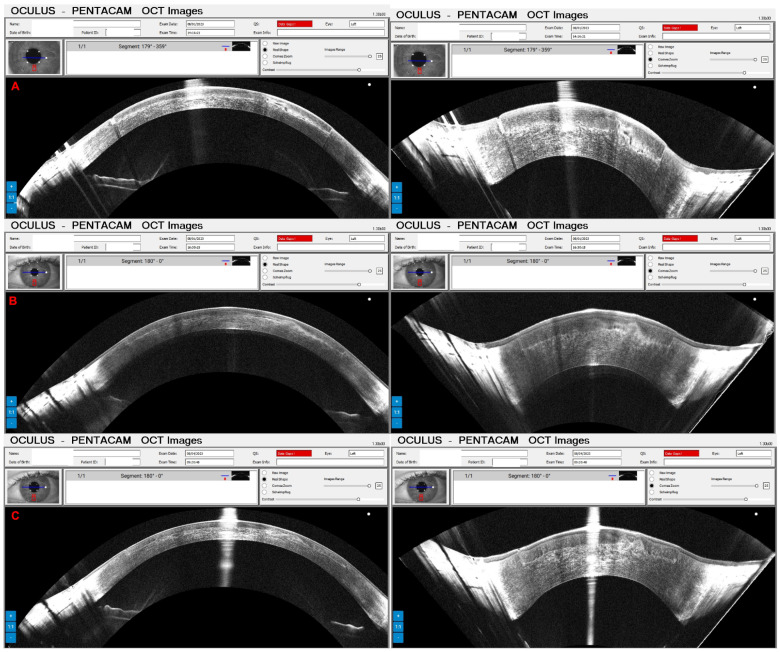
The OS Pentacam OCT of the same patient and periods as Figure 12. In (**A**), at the first exam, the PIO of 44 and 45 mmHg in OD and OS showed a curved cornea with edema. In (**B**), after two hours of administering carbonic anhydrase inhibitor orally, we can observe the corneal changes, which deflate and flattens; in (**C**), three days later, with PIO of 12 mmHg in both eyes and no pathological corneal changes.

**Figure 14 diagnostics-14-01455-f014:**
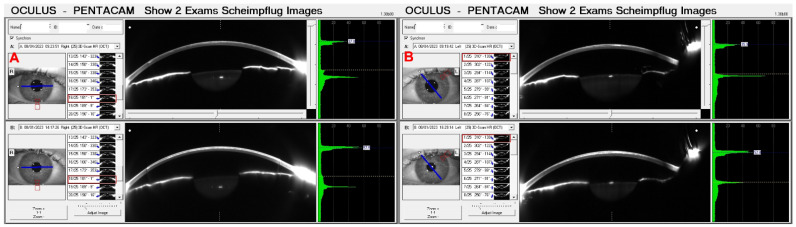
The 2 exams Scheimpflug images of the same patient as Figure 12 and Figure 13. We can observe the improvement of cornea edema three days after the treatment in OD (**A**) and OS (**B**).

**Figure 15 diagnostics-14-01455-f015:**
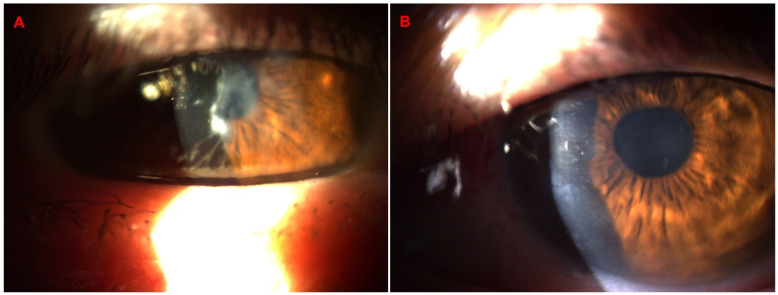
Slit-lamp biomicroscopy of a Salzmann nodular degeneration after a pterygium excision surgery 45 years ago (**A**) and one day after manual removal with PTK smoothing and mitomycin use (**B**).

**Figure 16 diagnostics-14-01455-f016:**
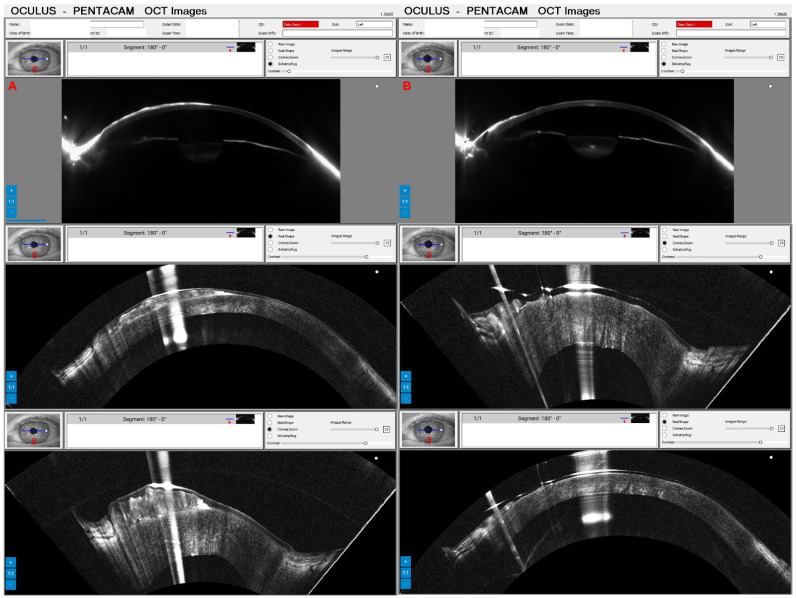
Pentacam OCT of the same patient as Figure 14 shows the Scheimpflug, real shape, and cornea zoom of the Salzmann nodular degeneration (**A**) and one day after manual removal with PTK smoothing and mitomycin use (**B**).

## Data Availability

All data generated or analyzed during this study are included in this published article.

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
