# Peer review of "Combined Rotating Ultra-High-Resolution Spectral Domain OCT and Scheimpflug Imaging for In Vivo Corneal Optical Biopsy"

_diagnostics, 2024, doi:10.3390/diagnostics14131455_

Round 1
Reviewer 1 Report
Comments and Suggestions for Authors
This paper is interesting since it is very practical. However, the section of discussion is too short. Authors should examine the section and are strongly recommended to describe more. Additionally, numbers of figures are too much and should clearly presented by photoshop etc.
Author Response
We appreciate the reviewers' insightful feedback and constructive comments on our manuscript titled "Combined Rotating Ultra-High-Resolution Spectral Domain OCT and Scheimpflug Imaging for in-vivo Corneal Optical Biopsy."
Reviewer 2 Report
Comments and Suggestions for Authors
General comments:
A shorter than usual review from me, but there are two related points that jumped out at me immediately.
Combined simultaneous OCT and Scheimpflug imaging has previously been published, which the authors have not addressed:
Li, Xiaoran, et al. "Simultaneous optical coherence tomography and Scheimpflug imaging using the same incident light." Optics Express 28.26 (2020): 39660-39676.
Y Shen, Y Zheng, S Lawman, X Li - US Patent 11,596,303, 2023
The full description of the Pentacam Cornea OCT hardware is not openly available to the reader yet, so add (optical) system diagram and full technical description of both OCT and Scheimpflug hardware here.
If both of these are addressed, then the content would be more than novel enough for an MDPI journal. However, without addressing both, my view on the work is negative. I therefore recommend major revisions, with the expectation of a second round of (probably minor) revisions when there is enough information for me to provide some more detailed points.
Specific comments:
Because the missing information from the two main points will affect how I interpret details, I am not going to provide specific comments at this stage.
One exception, “Jules Carpentier in 1901” and “Theodor Scheimpflug in his 1904 patent” need to be in the reference list.
Author Response
Dear Editors,
We appreciate the reviewers' insightful feedback and constructive comments on our manuscript titled "Combined Rotating Ultra-High-Resolution Spectral Domain OCT and Scheimpflug Imaging for in-vivo Corneal Optical Biopsy."
We are pleased to inform you that we have thoroughly addressed all the reviewers' concerns. Specifically:
- Literature Integration: We have included a detailed discussion of the previous work on combined OCT and Scheimpflug imaging, referencing the studies by Li et al. (2020) and Shen et al. (2023).
- Technical Description and Diagrams: We have added a comprehensive technical description of the OCT and Scheimpflug hardware used in the Pentacam Corneal OCT, including an optical system diagram.
- Historical References: The historical references to "Jules Carpentier in 1901" and "Theodor Scheimpflug in his 1904 patent" are referenced as 22, and 23.
Carpentier J. Appareil de projection cinématographique. French Patent No. 324,823. Paris: National Institute of Industrial Property; 1901.
Scheimpflug T. Verfahren und Vorrichtung zur Herstellung von photographischen Bildern und Karten. Austrian Patent No. 69237; 1904.
- We added a new co-author who actively participated in revising the article.
The Word document was used to have all highlighted tracked changes, including the figures.
These revisions significantly enhance the manuscript and address the reviewers' valuable suggestions. We look forward to the next steps in the review process and remain available for any further questions or clarifications.
I appreciate your consideration.
Yours sincerely,
Renato Ambrósio Jr, MD, PhD, FWCRS, PCEO
Round 2
Reviewer 2 Report
Comments and Suggestions for Authors
Minor revision – The new figure 1 and figure 2 are missing from the re-submission.
The questions I had have been answered. I think the overview of the system and example results are novel in the literature presently, and they would be of significant interest. Therefore I am happy to recommend the manuscript to be published.
Author Response
Dear Editor, We are glad to have this work accepted. The Figures were sent as a Ziped File. I will upload it again. Thank you. Sicnerely, Renato Ambrósio